# Analysis of Telomere Length and Its Implication in Neurocognitive Functions in Elderly Women

**DOI:** 10.3390/jcm11061728

**Published:** 2022-03-21

**Authors:** Juan Luis Sánchez-González, Juan Luis Sánchez-Rodríguez, Raúl Juárez-Vela, Regina Ruiz de Viñaspre-Hernandez, Rogelio González-Sarmiento, Francisco Javier Martin-Vallejo

**Affiliations:** 1Department of Nursery and Physiotherapy, Faculty of Nursery and Physiotherapy, University of Salamanca, 37007 Salamanca, Spain; juanluissanchez@usal.es; 2Department of Basic Psychology, Psychobiology and Methodology, Faculty of Psychology, University of Salamanca, 37005 Salamanca, Spain; 3Research Group in Care, Department of Nursing, University of La Rioja, 26002 Logroño, Spain; raul.juarez@unirioja.es; 4Department of Medicine, Faculty of Medicine, University of Salamanca, 37007 Salamanca, Spain; gonzalez@usal.es; 5Department of Statistics, Faculty of Medicine, University of Salamanca, 37007 Salamanca, Spain; jmv@usal.es

**Keywords:** telomere, cognition, adults, aging

## Abstract

During the normal aging process, a series of events occur, such as a decrease in telomere length and a decrease in various cognitive functions, such as attention, memory, or perceptual-motor speed. Several studies have attempted to establish a correlation between both variables; however, there is considerable controversy in the scientific literature. The aim of our study was to establish a correlation between the scores obtained in the following different cognitive tests: the Mini-Mental State Examination, the Benton Visual Retention Test, the Trail Making Test, the Rey Auditory Verbal Learning Test, the Stroop Test, and the measurement of telomere length. The sample consisted of a total of 41 physically active, healthy women, with a mean age of 71.21 (±4.32) and of 33 physically inactive, healthy women, with a mean age of 72.70 (±4.13). Our results indicate that there is no correlation between the scores obtained by the women in either group and their telomere length. Therefore, it is not possible to conclude that telomere length can be correlated with cognitive performance.

## 1. Introduction

The aging of the population brings with it a significant increase in age-related diseases, causing a high degree of disability and dependence in the people who suffer from them [1]. Cognitive impairment (CI) of the population begins in adulthood [2] and continues progressively. It is estimated that the main causes of CI are environmental factors and, above all, genetic factors [3]; however, the exact cause of CI in the population is not known. One of the most refuted theories about the aging process is the theory of telomeres and the enzyme telomerase. Telomeres are cellular structures composed of tandem DNA repeats (TTAGGG) located at the end of chromosomes, and their main function is to protect chromosomes from degradation during each cell cycle. The enzyme responsible for the maintenance of telomeric length is telomerase [4]. The activation of telomerase can treat diseases related to aging and prolong human life, as published by de Jesús and Blasco [5]. It is estimated that approximately 71–72 base pairs are lost per year in both male and female populations [6]. Due to this telomere shortening and its association with the processes of apoptosis and cell senescence, telomeres have been proposed as the main indicators of cell aging [7]. There are different factors that favor greater telomeric attrition, such as neoplastic processes, smoking, diabetes, and cardiovascular diseases [8,9,10]. However, there is also a series of factors that help to protect this telomeric wasting. One of the most studied factors, in terms of its implication on telomere length (TL), is physical exercise. Several independent investigations have established connections between TL and physical exercise, or general fitness. A study of 582 participants in the Cardiovascular Health Study revealed a positive association between the distance traveled and the telomere length [11]. A recent review concluded that aerobic and endurance physical exercise prevents telomeric shortening and DNA damage [12].

Several studies have examined the relationships between TL and cognitive functioning in people without dementia, although the evidence to date is equivocal. Some of these studies have found relationships between longer telomere length and better performance in global cognitive function, assessed through the Mini-Mental State Examination (MMSE) [13,14,15], while others have not been able to establish a clear correlation [16,17].

This great variability in results could be due to the different methodologies used by the studies, as well as the different populations, different cognitive functions evaluated, different tests used, and different laboratory protocols used to obtain the TL. In addition, another aspect to take into account is the sociodemographic differences in the samples studied. Some studies affirm that the association between TL and cognition can only be found in socially disadvantaged groups [18,19]. Similarly, another factor to take into account in this possible relationship is the regular practice of physical exercise by the study populations. In previous studies, we have confirmed that physical exercise has a positive impact on both telomere length and cognition [20,21]. However, as we have previously mentioned, there is much controversy regarding the possible relationship between TL and cognitive function.

One of the possible hypotheses by which TL could be related to cognition lies in the genetic component. Novel studies have shown that telomere length has a strong inherited genetic component (between 34 and 82%), as stated by Broehr et al. [22]. Specifically, up to 16 inherited genetic variants are known [23]. This possible mechanism could explain why people with degenerative diseases, with a certain inherited genetic component (e.g., Alzheimer’s disease, Parkinson’s disease, and frontotemporal dementia) and high cognitive impairment, have a shorter telomere length.

In 2016, a study was published that stated that people with Alzheimer’s disease present an altered profile of the base excision DNA repair system in their brain tissue and blood, presenting an alteration in the expression of the BER gene [24]. Perhaps this gene could also be implicated in excessive shortening of the TL, due to an alteration in the DNA base excision repair mechanisms. The present study aimed to investigate the relationship between TL and cognition in physically active and physically inactive people.

## 2. Materials and Methods

### 2.1. Design and Population

A quasi-experimental intervention design was conducted. A convenience sample of 74 patients aged 71.21 (±4.32) in the inactive group and 72.70 (±4.13) in the active group were recruited to participate in this study. All study participants were informed of the details and considerations of the research, and subsequently signed an informed consent form in order to participate in the research. The study was approved by the Bioethics Committee of the University of Salamanca. Anonymity and protection of personal data was guaranteed by providing an identification number to each participant.

The subjects of the physically active group were recruited from the Geriatric Revitalization Program (GRP), run by the Faculty of Nursing and Physiotherapy, at the University of Salamanca; the subjects of the non-physical activity group were selected from programs run by the Salamanca City Council (Figure 1).

The inclusion criteria were as follows: being a woman, not taking medication for blood pressure, not having diabetes, not smoking, not having neoplastic processes or neurodegenerative diseases, and being between the ages of 65 and 85.

Active subjects participated in the GRP, performing aerobic and strength resistance physical exercise 3 days a week for a total of 6 months. The basic GRP session was divided into the following 3 distinct parts:

The first part consisted of a warm-up, including active mobility of the main joints, along with 5–10 min of aerobic exercise. 

The second part, called the “main part”, consisted of strength resistance exercise, involving the main muscle groups of the upper extremities (biceps, triceps, and deltoids), the trunk (chest, back, and abdominals), and the lower extremities (rectus femoris and gastrocnemius). This main part lasted about 25–30 min. Once the main part was finished, a brief hydration period was performed.Finally, there was a cool down session, followed by breathing exercises, and by stretching the main muscle groups involved in the session.

Participants in the inactive group did not participate in any type of activity involving regular physical exercise or moderate-intensity exercise (>30 min per day) for more than 3 days per week, and were, therefore, defined as “currently inactive” [25,26].

### 2.2. Variables 

#### 2.2.1. Telomere Measurements

The following protocol was followed for the extraction of DNA from saliva [27,28] and the same laboratory steps were followed as in previous studies [21]. The cells were isolated by centrifugation and resuspended in Fornace buffer (0.25 M sucrose, 50 mM Tris-HCl pH 7.5, 25 mM KCl, and 5 mM MgCl2), followed by additional centrifugation at 1500 rpm for 10 min. The resulting pellet was incubated at 55 °C for 8–16 h in Fornace buffer, containing 10 mM EDTA pH 8.0, 1% SDS, and proteinase K (ApliChem, Castellar del Vallès, Spain), for protein degradation and for breaking down the cell membrane. Then, DNA was extracted and purified using the phenol–chloroform method and precipitated using cold absolute ethanol.

The concentration of the extracted DNA was determined by measuring the absorbance at 260 nm using a NanoDrop ^TM^ 2000/2001 spectrophotometer. The purity of the DNA was analyzed based on the A260/280 absorbency ratio, where an optimal purity ratio ranged between 1.8 and 2.0.

The telomere length of the saliva cells taken from each participant was measured using quantitative real-time PCR (qPCR) together with the Absolute Human Telomere Length Quantification qPCR Assay Kit (ScienCell, Catalog #8918, Faraday Ave, Carlsbad, CA, USA).

This technique allows the initial amount of DNA coding for telomerase (TEL) to be quantified and compared with that obtained simultaneously from another fragment corresponding to a single copy reference gene (SCR) exerting endogenous control. The difference in the amount of DNA quantified represents the relative TL of each participant. In order to analyze these relative changes, a reference fragment (CONTROL) of known TL (provided by the manufacturer) was added to each assay, allowing absolute quantification of the telomere length of each sample.

The reactions took place in a Micro-Amp Fast Optical 96-Well reaction plate (Applied Biosystems), and the TEL and SCR fragments from the DNA samples of participants were amplified using the Applied Biosystems StepOnePlusTM Real-Time PCR System. Triplicate reactions were carried out for each sample to minimize variability. The TEL and SCR fragments were amplified using 1 uL of each specific primer, 1 uL of the FastStart SYBR Green MasterMix, and 7 uL of water. The total amount of DNA used for each reaction was 10 ng in 2 uL. The amplification program was as follows: 10 min at 95 °C, followed by 40 cycles at 95 °C for 15 s, 52 °C for 30 s and 60 °C for 1 min.

Finally, the Ct [2^−ΔΔCt^] comparative method was used to calculate the relative expression levels of each amplicon. The specificity of each PCR was checked by verifying the TL of the reference sample (CONTROL), which, in turn, allowed the absolute lengths of each sample for a diploid cell and/or chromosome end to be determined.

Telomere length measurements were taken at the same time in both the active and inactive groups.

#### 2.2.2. Cognitive Functions Assessments

We evaluated cognitive function using the Mini-Mental State Examination (MMSE) [29], Benton’s Visual Retention Test (BVRT) [30], Rey’s Audio Verbal Learning Test (RAVLT) [31], the Stroop Test [32], and Trail Making Test (TMT) [33]. 

The neuropsychological evaluation was performed in a single moment in both groups, as was the case with the TL.

### 2.3. Statistical Procedures

Quantitative variables data were analyzed using means and standard deviations. The percentages were analyzed for the qualitative variables. The t-test was used to compare the independent data of both groups when the variables followed a normal distribution. Levene’s test was used to assess the equality of variances of the populations, and to calculate the degrees of freedom for the t-test. Pearson’s correlation coefficient [r] was calculated to explore the association between telomere length and cognitive function. Given the importance of weight and body fat (%) on the length of telomeres, partial correlation [rp] analysis was also carried out between telomere length and cognitive function to control these variables. The t-test was used to test the significance of the correlation coeefficient. The correlation coefficients were compared with the Z-test using Fisher’s r-to-z transformation. Benjamini–Hochberg adjustment was applied, and the significance level 0.05 was chosen. SAS-JMP v 12 and R statistics were used to analyze the data.

## 3. Results

The results of the comparisons between the sociodemographic variables show no statistically significant differences between the active and non-active groups (Table 1). While significant differences have been detected in the weight between the active and non-active groups (t = 3.68; df = 72, *p*-value = 0.004), the percentage of fat was not statistically significant (t = 1.79; df = 72; *p*-value = 0.381). Table 1 lists the descriptive statistics by age group, years of schooling, marital status and educational level. 

The magnitude of correlations between cognitive functions and telomere length is weak and is similar in both groups (Figure 2). Only a moderate correlation is detected in the BVRT true (r = −0.37) in the inactive group, although it is negative. If the correlation coefficients corrected for weight and percentage of fat (partial correlations) are inspected (Table 2), the strength of the correlation coefficients is similar to the uncorrected coefficients, although with slight increases in magnitude in most of the associations. All the correlation coefficients in both groups are not significantly different from zero. Trail Making A and B show higher and negative partial correlation coefficients in the active compared with the inactive group, although there are no statistically significant differences detected between the correlation coefficients of both groups.

## 4. Discussion 

This study’s central finding suggests that there is no significant correlation between TL and neurocognitive test scores in both physically active and physically inactive women. Previous studies examining the relations between TL and cognition have reported equivocal findings. Although some studies found associations between longer TL and better performance on global cognitive screening measures [13,14,34], as well as specific cognitive domains [35], there were others that found null associations [9,11,12,27,28,29,30]. Our results are in line with those published by authors such as Kaja et al. [36,37,38,39] where they examined the relationship between LT and cognitive performance in a sample of more than 2000 subjects. Cognitive capacity was evaluated through the MoCA Test; it was concluded that there was no statistically significant relationship between telomere length and the losses obtained in said test. Authors such as Hagg et al. [40] obtained similar results by analyzing the observational and causal associations between telomere length and the following five cognitive variables: general cognitive function, processing speed, visuospatial functioning, memory, and executive functioning, in 12 population cohorts, observing an association between the LT and the scores obtained in cognitive performance, although they did not reach levels of statistical significance. As we mentioned in the Results section, Trail Making A and B showed higher and negative partial correlation coefficients in the active group compared with the inactive group, although there were no statistically significant differences detected between the correlation coefficients of both groups. We have only found one study [18] that concluded that those people who showed a larger telomere size showed a better performance in the Trail Making Test B, although it is true that the population sample was different to that used in our study. The explanation for such disparate results could be that there are many confounding factors associated with the study populations, which could directly influence cognition and telomere length. One of these confounding factors could be genetics [36,41]; however, studies have also found significant differences in cognitive scores between twins with inconsistent DL, and the results of these studies confirmed that the observed correlation between decreased cognitive ability and shortened TL is robust, despite age and potential confounders. Similarly, inflammation and oxidative stress could have a direct influence on TL and, therefore, on cognition, since telomeres are highly sensitive to damage by oxidative stress and inflammation [42]. Another of the confounding factors associated with TLand cognition is sociodemographic variables. Leibel and colleagues [18] concluded that TL was associated with poor performance in tests of executive function among participants living in poverty, particularly Caucasians. A possible explanation for these results is that people living in poverty are exposed to a greater number of biopsychosocial risk factors, such as cardiovascular risk, diseases, and chronic stress [43,44].

Therefore, it is clear that all these confounding factors must be taken into account when analyzing the correlations between TL and cognition. If there is no exhaustive control over the study variables, it is likely that the same results will continue to be obtained. An important focus for future research will be to explore all of the confounding factors that could explain the association between TL and cognition.

Our study has a number of strengths and limitations. Regarding the strengths, it should be noted that the present study included a wide battery of neuropsychological tests, so a large number of cognitive functions were addressed. In addition, the two study groups, physically active women and physically inactive women, were very homogeneous, in terms of age, educational level, weight, fat percentage, and the number of people in each group. In addition, other factors should be taken into consideration, as only having a sample of women may have introduced gender bias. It will be necessary for future researchers to know the degree of poverty, the genetics, and the occupation of the populations under study. It will also be beneficial to increase the number of participants to obtain results that are more consistent and robust, and to control the full randomization of the study. Thus, this study is classified as quasi-experimental.

## 5. Conclusions

No evidence has been found that suggests a direct relationship between cognitive variable scores and telomere length in physically active and physically inactive subjects. A greater number of studies, samples and more homogeneous methodologies are needed to demonstrate that telomere length is a variable that is directly related to cognition.

## Figures and Tables

**Figure 1 jcm-11-01728-f001:**
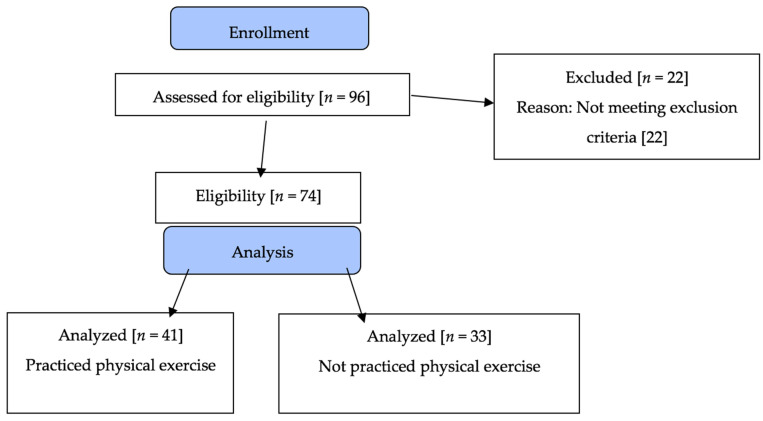
Flow Chart.

**Figure 2 jcm-11-01728-f002:**
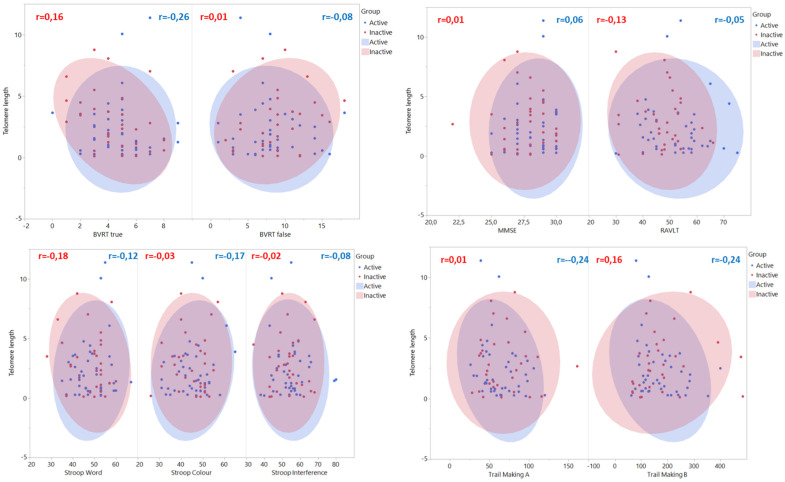
Scatterplot and correlation coefficients [r] from cognitive functions and length of telomere. The 95% bivariate normal density ellipse is shown in each scatterplot. This ellipse encloses approximately 95% of the points.

**Table 1 jcm-11-01728-t001:** Sociodemographic variables.

Sociodemographic Variables	Inactive Group Mean [SD]/Counts				Active Group Mean [SD]/Counts				*p*-Value
Age	71.21 (±4.32)				72.70 (±4.13)				0.138
Years of schooling	8.42 (±2.56)				8.18 (±1.55)				0.576
Marital Status	Single	Married		Widower	Single	Married		Widower	0.238
	9 (27.3%)	19 (57.6%)		5 (15.2%)	6 (14.6%)	30 (73.2%)		5 (12.25%)	
Educational level	Primary/G.B.E 21 (63.6%)		Mid-Higher level 12 (36.4%)		Primary/G.B.E 20 (8.48)		Mid-Higher level 21 (51.2%)		0.201

Note. SD = standard deviation; G.B.E = general basic education.

**Table 2 jcm-11-01728-t002:** Partial correlation coefficients (adjusted statistical significance) and adjusted p-value from the comparison of coefficient correlation between groups.

	Inactive Group (r_p_)	Active Group (r_p_)	Adjusted *p*-Values
MMSE	0.015 (n.s)	0.057 (n.s)	0.853
RAVLT	−0.132 (n.s)	−0.022 (n.s)	0.731
BVRT true	−0.404 (n.s)	0.078 (n.s)	0.176
BVRT false	0.237 (n.s)	−0.153 (n.s)	0.315
Stroop word	−0.196 (n.s)	0.119 (n.s)	0.374
Stroop colour	0.014 (n.s)	0.172 (n.s)	0.731
Stroop interference	0.015 (n.s)	−0.097 (n.s)	0.731
Trail Making B	0.194 (n.s)	−0.299 (n.s)	0.176
Trail Making A	0.026 (n.s)	−0.274 (n.s)	0.374

Note. MMSE = Mini-Mental State Examination; BVRT = Benton Visual Retention Test; RAVLT = Rey Auditory Verbal Learning Test.

## Data Availability

The datasets analyzed during the current study are available from the corresponding author upon reasonable request.

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
