# Peer review of "Analysis of Telomere Length and Its Implication in Neurocognitive Functions in Elderly Women"

_jcm, 2022, doi:10.3390/jcm11061728_

Round 1
Reviewer 1 Report
The manuscript of Sánchez-González et al. tests correlation between telomere length and cognitive functions. The manuscript is in general very sloppy, contains typing errors, mistakes in English, lost words forgotten in the text, not explained abbreviations, etc. It presents negative results that may not be of high (if any) scientific interest.
The introduction is too short and lacks a proper explanation of the objectives of the study. Has a hypothesis describing a possible molecular mechanism through which telomere length would influence cognitive functions ever been described in literature?
The study design is not properly described either, e.g. why was the study performed on female patients only?
The methods section contains multiple mistakes and lay formulations:
- it is not necessary to mention that DNA was stored in Eppendorf tubes ...
- was really "amount of DNA coding for TELOMERASE" quantified?
- primer concentration is not mentioned, microlitres of water do not have to be mentioned ...
- what type of t-test was used?
Results are shortly presented in a form of two tables and one figure of a very poor quality. Limitations of the study are mentioned twice - first in discussion, second in a separate section "Limitations".
Author Response
Dear Reviewer, thank you for all your comments. We have addressed it.
Yours Faithfully.
Dr. Juárez-Vela on behalf of all authors.

Reviewer 2 Report
The title should change to "Analysis of Telomere Length and its Implication in Neurocognitive Functions IN ELDERY WOMEN".
In the Introduction - please correct the repeat sequence - TTAGGG
In the Discussion - "future research to consider the degree of poverty or the genetics", also consider the. occupation. Depending on the occupation, the telomeres may shorten.
Author Response
The title should change to "Analysis of Telomere Length and its Implication in Neurocognitive Functions IN ELDERY WOMEN".
Thanks so much. We adressed it .
In the Introduction - please correct the repeat sequence – TTAGGG
Thanks so much. We adressed it .
In the Discussion - "future research to consider the degree of poverty or the genetics", also consider the. occupation. Depending on the occupation, the telomeres may shorten.
Thanks so much. We addressed it.
Dear Reviewer thanks so much for your review. We appreciate all the changes.
Round 2
Reviewer 1 Report
The authors have clearly improved the quality of the manuscript's text and figure. However, I still miss sufficient theoretical introduction, especially explaining hypotheses for the relation between telomere length and cognitive functions, e.g. is accelerated telomere shortening in theory a cause or a result of the age-related disease (?), eventually describing related molecular mechanisms, e.g. is it only the shorter lifespan of the respective cells speculated to play role in the disease or also misbalance in signalling pathways that are triggered upon telomeres erosion (?).
Regarding the description of the composition of the qPCR reaction, mentioning only volume and not concentration of the primer is really insufficient.
Author Response
First of all, thank you very much for your comments, we are very pleased that you consider that we have improved the clarity of the text.
Regarding the hypothesis of telomere length and cognitive functions. It has been studied that telomeres have the function of protecting the end of chromosomes avoiding genomic instability. They shorten with each cell cycle contributing to replicative senescence when they reach the so-called "Hayflick limit". There is an enzyme that replenishes the loss of these telomeres during replication, this enzyme is known as telomerase. The activation of telomerase can treat diseases related to aging and prolong human life as published by de Jesús & Blasco, 2013.
Humans can age in two ways: on the one hand physiological aging in which cognitive functions, physiological and functional changes do not impact the quality of life. However, another group of people may experience pathological aging in which cognitive, physiological, and functional functions may have an impact on quality of life. The cognitive functions most affected in this pathological aging include memory, visuospatial skills, executive functions, attention, and perceptual-motor speed.
In this situation, it is hypothesized (not yet proven) that those people who age more rapidly and with repercussions on their quality of life (decrease in cognitive functions) could be associated with a greater shortening of telomere length or, on the contrary, that this telomere shortening is influenced by a decrease in cognitive functions.
One of the possible explanations could be because telomere length has a strong hereditary genetic component (from 34 to 82%) as stated by Broehr et al 2013. Specifically, up to 16 inherited genetic variants are known (Haycock et al, 2017). This possible mechanism could explain why people with degenerative diseases with a certain hereditary genetic component (Alzheimer's disease, Parkinson's disease, frontotemporal dementia) with a high cognitive impairment have a smaller telomere size.
However, it would be interesting to know if the DNA repair mechanisms (direct and indirect) involved in genomic stability are also related to the telomere length of people with alterations in cognitive functions, i.e., those people with alterations in cognitive functions have a greater number of errors or alterations in DNA repair mechanisms. We believe that these are lines of research that have yet to become evident and it would be interesting to continue in this direction
References :
Broer, L. , Codd, V. , Nyholt, D. R. , et al. (2013). Meta‐analysis of telomere length in 19,713 subjects reveals high heritability, stronger maternal inheritance and a paternal age effect. European Journal of Human Genetics, 21(10), 1163–1168. 10.1038/ejhg.2012.303
de Jesús, B. B. , & Blasco, M. A. (2013). Telomerase at the intersection of cancer and aging. Trends in Genetics, 29(9), 513–520. 10.1016/j.tig.2013.06.007
Haycock, P. C. , Burgess, S. , Nounu, A. , Zheng, J. , Okoli, G. N. , Bowden, J. , … Davey Smith, G. (2017). Association between telomere length and risk of cancer and non‐neoplastic diseases: A mendelian randomization study. JAMA Oncology, 3(5), 636–651. 10.1001/jamaoncol.2016.5945
Regarding the description of the composition of the qPCR reaction, mentioning only volume and not the concentration of the primer is really insufficient.
Unfortunately, the concentration of the primer is not available. The experiments are performed with a commercial kit: Absolute Human Telomere Length Quantification qPCR Assay Kit (AHTLQ) Catalog #8918. and the laboratory does not share it.
Thanks for all your consideration we appreciate it so much!